# Invasive candidiasis following lung transplant: An Assessment of impact utilizing a national insurance claims cohort

Kelly M. Pennington[1,2*‡], Herb Heien[3,4,‡], Hemang Yadav[1,2], Xiaoxi Yao[3,4,5], Bradley White[1], Steve G. Peters[1,2], Patricio Escalante[1], Che Ngufor[3], Raymund R. Razonable[2,6], Cassie C. Kennedy[1,2,3]

1 Division of Pulmonary and Critical Care Medicine, Department of Medicine, Mayo Clinic Rochester, Rochester, Minnesota, United States of America, 2 William J. von Liebig Center for Transplantation and Clinical Regeneration, Mayo Clinic Rochester, Rochester, Minnesota, United States of America, 3 Robert D. and Patricia E. Kern Center for the Science of Healthcare Delivery, Mayo Clinic Rochester, Minnesota, United States of America, 4 OptumLabs Eden, Eden Prairie, Minnesota, United States of America, 5 Department of Cardiovascular Medicine, Mayo Clinic, Rochester, Minnesota, United States of America, 6 Division of Infectious Diseases, Department of Medicine, Mayo Clinic Rochester, Rochester, Minnesota, United States of America

‡These authors had equal contributions to the study and manuscript development. They should be considered co-first authors
* Pennington.Kelly@mayo.edu

## Abstract

### Introduction

Lung transplant recipients (LTRs) are susceptible to invasive candidiasis (IC). This study aimed to assess the incidence, risk factors, and impact of IC on mortality in LTRs using a national insurance claims cohort.

### Methods

We conducted a retrospective cohort study using administrative claims data from the OptumLabs® Data Warehouse. We identified LTRs from January 1, 2005, to December 31, 2023, using procedural codes. Exclusion criteria included re-transplantations and pre-transplant IC. We employed multivariable logistic regression to identify risk factors for IC and Cox Proportional Hazard models to assess the impact of IC on mortality.

### Results

Among 1279 LTRs, 131 (10.2%) developed IC, primarily during the initial hospitalization for lung transplantation (index hospitalization). The median time to IC diagnosis was 32.0 days following transplant. Post-transplant extra-corporeal membrane oxygenation (ECMO) for more than 8 days was associated with IC (OR: 2.34; 95% CI 1.03 to 5.34). Mortality was higher in LTRs with IC (HR: 2.31; 95% CI: 1.45 to 3.67;

**Data availability statement:** Data Sharing Statement: This study was conducted using de-identified administrative claims from OptumLabs Data Warehouse. These data are third party data owned by OptumLabs and contain sensitive patient information; therefore, the data is only available upon request. Interested researchers engaged in HIPAA compliant research may contact connected@optum.com for data access requests. The data use requires researchers to pay for rights to use and access the data. These data are subject to restrictions on sharing as a condition of access.

**Funding:** The author(s) received no specific funding for this work.

**Competing interests:** The authors have declared that no competing interests exist.

**Abbreviations:** COPD, Chronic obstructive pulmonary disease; C.I., Confidence interval; PH, Cox proportional hazard model; CPT Current procedural terminology; CMV, Cytomegalovirus; ECMO, Extra-corporeal membrane oxygenation; IPF, Idiopathic pulmonary fibrosis; ICD-9, International classification of diseases, ninth revision; ICD-10, International statistical classification of diseases and related health problems, tenth revision; I.C., invasive candidiasis; IFI, Invasive fungal infection; ILD, Interstitial lung disease; L.T., Lung transplant; LTR, Lung transplant recipient; O.R., Odds ratio; OLDW, OptumLabs® data warehouse; US, United States.

p < 0.001). LTRs with IC also had longer hospital stays (median 26.0 days vs. 20.0 days; p < 0.001) and more re-operations (36.7% vs. 27.3%; p = 0.003) compared to those without IC.

## Conclusion

Invasive candidiasis affects approximately 10% of lung transplant recipients, most often during the initial hospitalization. It is associated with increased mortality, prolonged hospital stays, and a greater need for surgical re-intervention. These findings highlight the importance of early identification and targeted preventive strategies to improve outcomes in this high-risk population.

## Introduction

Lung transplant recipients (LTRs) are particularly vulnerable to invasive fungal infections (IFIs), which occur in up to 25% of recipients and are linked to an alarming three-fold increase in all-cause mortality [1–3]. While *Aspergillus spp.* have traditionally been seen as the primary cause of IFI in LTRs [4–6], more recent single-center studies suggest that invasive candidiasis (IC) may also be an important cause of IFI, particularly within the first 90 days post-lung transplant period [7,8]. Candida spp. possess virulence factors that promote adhesion to host tissues, hyphal transformation, and enzymatic degradation of epithelial barriers, allowing for tissue invasion and dissemination, particularly in immunocompromised hosts [9].

While many lung transplant centers in the United States employ antifungal prophylaxis targeting mold infections [10], variation exists in the agents used and the duration of prophylaxis. Some studies have suggested that the use of inhaled amphotericin B without systemic antifungal prophylaxis may be insufficient to prevent all forms of invasive fungal infection, including invasive candidiasis [7]. In the absence of universal prophylaxis, the incidence of candidemia at a single Canadian transplant center was reported at 3.5% with most episodes occurring within the first 30 days post-transplant [8]. Risk factors for candidemia included pre-transplant hospitalization, post-transplant extracorporeal membrane oxygenation (ECMO), and post-transplant renal replacement therapy [8]. In non-LT populations, the risk factors for IC and candidemia are clearly established and additionally include gastrointestinal perforation, length of hospital stay, and diabetes mellitus [11].

While universal antifungal prophylaxis may seem like an effective preventative measure for IC, evidence in the lung transplant population remains limited and inconclusive. Moreover, antifungal medications can have significant adverse effects, with reported rates including QTc prolongation in up to 20% of patients, hepatotoxicity in 5–15%, periostitis in approximately 10% of those receiving long-term voriconazole, and increased risk of skin cancer with prolonged triazole exposure [12,13]. Triazole antifungal medications are also expensive and interact with immunosuppressive agents, specifically calcineurin inhibitors, which have a narrow therapeutic index necessitating close, serial monitoring.

Given the unclear benefit, potential for toxicity, excessive costs, and drug interactions of antifungal prophylactic medications, it is imperative to have a clear understanding of the true impact of IC and the risk factors associated with IC in LTRs. Large claims databases, such as OptumLabs® Data Warehouse (OLDW), have been widely used to assess disease incidence and risk factors across various clinical domains [14,15]. The ability to identify a large Mult institutional cohort of LTRs, control for potential confounders, identify risk factors and diagnostics, all while following the cohort longitudinally makes this approach particularly attractive in attempting to identify the true impact of IC in LTRs.

This study aimed to describe the incidence and risk factors associated with IC in LTR and assess mortality among LTRs with IC and without IC using individuals enrolled in commercial and Medicare Advantage health plans in the US.

## Methods

This study was conducted in accordance with the STROBE (Strengthening the Reporting of Observational Studies in Epidemiology) guidelines. A detailed STROBE checklist has been included as a supplementary document to ensure comprehensive and transparent reporting of the research methods and findings.

### Data source

We conducted a retrospective cohort study using de-identified administrative claims data from OLDW, which includes medical and pharmacy claims and enrollment records for commercial and Medicare Advantage enrollees. This database contains longitudinal health information on enrollees and patients from a diverse mixture of ages, ethnicities and geographical regions across the US [16]. OLDW provides real-world outcomes data for LTR in geographically and demographically diverse populations. In accordance with the Health Insurance Portability and Accountability Act, the use of pre-existing, de-identified claims data is exempted from Institutional Review Board review.

### Study population

We extracted all single or bilateral lung transplants occurring in adults (≥ 18 years of age) from January 1, 2005 to November 30, 2023 using eligible *International Classification of Diseases, Ninth Revision* (ICD-9); *International Statistical Classification of Diseases and Related Health Problems, Tenth Revision* (ICD-10); and Current Procedural Terminology (CPT) codes as previously described [17]. We required at least 90 days of continuous enrollment prior to transplant to allow for adequate baseline characterization, and at least 30 days post-transplant to ensure sufficient follow-up to assess early post-transplant outcomes. Follow-up started after the patients' index date and continued to the end of enrollment, death, or end of the study period (December 31, 2023).

Re-transplantations were excluded by removing patients who had multiple hospital encounters with LT procedure codes. Re-transplantation cases were excluded to reduce heterogeneity, as these patients often have distinct clinical characteristics and risk profiles compared to primary transplant recipients. Patients were required to have a LT stay of at least 5 or more days, and those who had same-day admission and discharge dates were not excluded as they were considered aborted transplant procedures, also known as "dry runs." Patients with IFI prior to their index date were also excluded to help reduce any potential confounding.

### Outcomes of interest: risk factors for developing invasive candidiasis and its effect on mortality

Our study aimed to evaluate both risk factors of IC, and its effect on mortality.

In our population, IC was defined as any diagnostic code for IC (ICD-9: 112.4, 112.5, 112.8, 112.83; ICD-10: B37.1, B37.5, B37.6, B37.7) in any position following the LT procedure. *These codes include candidemia (e.g., 112.5, B37.7), disseminated candidiasis (e.g., 112.5, 112.83), and organ-specific invasive infections (e.g., B37.5 for candidal peritonitis and B37.6 for candidal endocarditis).). We acknowledge that claims data may not distinguish between confirmed invasive*

*disease and coding inaccuracies.* These codes could occur during a clinic encounter, emergency department visit, or inpatient stay. Patients with IFI prior to the admission date for LT were excluded from the analysis. Mortality data were primarily obtained from the Social Security Death Master File, a national database that tracks death records. To increase completeness, particularly for deaths occurring during hospitalization, this was supplemented with information from hospital discharge status codes and insurance disenrollment records explicitly attributed to death. This approach has been validated in prior studies using the OptumLabs Data Warehouse [17,18].

Independent variables of interest included sex, age, race/ethnicity, census region, single versus double lung transplant, and indication for transplant. Elixhauser comorbidity score was calculated using ICD-9 and ICD-10 diagnostic codes captured during the 90-day baseline period prior to transplant, applying the Quan et al. adaptation of the Elixhauser algorithm [19]. The Elixhauser comorbidity index was dichotomized at ≥4 to define high comorbidity burden, consistent with distributional patterns in our data and previous literature. ECMO duration was dichotomized at >8 days based on prior studies linking prolonged (>7 days) ECMO support to increased post-transplant complications [20]. Baseline factors were extracted during the respective 90-day baseline period from the index date. An algorithm to identify the indication for transplant utilized ICD-9 and ICD-10 codes aimed at the specific chronic respiratory diseases. These codes could be found in any diagnosis position on a claim during the lung transplant stay to classify the most likely indication for transplant [17].

Other factors considered were pre-transplant corticosteroid use, post-transplant re-operation within the lung transplant hospital stay, post-transplant extra-corporeal membrane oxygenation (ECMO), post-transplant renal replacement therapy, and post-transplant cytomegalovirus (CMV) disease. Pre-transplant corticosteroid use was defined as a pharmacy prescription fill for prednisone, dexamethasone, or equivalent within 30 days prior to the lung transplant procedure. Post-transplant re-operation was defined as a procedure code with description of "chest closure", "chest exploration" or "chest washout" (CPT: 00520, 21750, 35820; ICD-9: 3451, 3452) occurring more than 24 hours after the initial transplant procedure and within the initial hospitalization for lung transplantation (index hospitalization). Post-transplant ECMO support and post-transplant renal replacement therapy were defined as ICD-9 or ICD-10 (ECMO: Z92.81; renal replacement therapy: Z99.2) diagnostic or procedure code within the index hospitalization. Post-transplant CMV disease was defined as the presence of an ICD-9 (771.1) or ICD-10 (B25.9) diagnostic code for CMV disease in any diagnostic position following transplant.

Antifungal medications prescribed within 90 days prior to a diagnostic code for IC were also considered in our analysis. Antifungal medications included caspofungin, micafungin, anidulafungin, fluconazole, itraconazole, posaconazole, voriconazole, isavuconazonium, amphotericin B deoxycholate, and liposomal amphotericin. Nebulized and intravenous amphotericin B deoxycholate and liposomal amphotericin could not be differentiated. In the absence of prior diagnostic codes for IFI, these were assumed to be nebulized.

## Statistical analysis

To protect patient confidentiality and in accordance with OptumLabs® restrictions, any event frequency of 11 or fewer participants was masked. Descriptive statistics were employed to describe the overall cohort. Categorical variables were summarized as frequency (%) and were compared using chi-square test. Continuous variables were expressed as median with interquartile range (IQR), as appropriate and were compared using Student's t test or the Wilcoxon rank-sum test, depending on distributional characteristics assessed by visual inspection and normality testing. The tables were analyzed across the IC and non-IC groups to test for differences.

Analysis of both risk factors in the development of IC and mortality from IC required us to use two different models and extraction methods. For both sets of analysis, we re-extracted patient enrollment and baseline characteristics and analyzed the data using both descriptive and multivariable methods. Each data extraction was anchored on each of the respective index dates, which created two very similar but unique sets of data for analysis. The patient counts appear approximately the same but should be thought of as two different data sets with very similar patient characteristics.

## Risk factor analysis

In our first analysis, we considered all LTRs who met the inclusion criteria for our study. We then assessed *a priori* which risk factors were associated with IC. The analysis was complicated by the fact that IC was most likely to occur during the LT stay, thus to ensure that all considered time dependent risk factors were measured prior to IC, we defined the index date as either the inpatient admission date or the first ECMO date after the LT transplant so that the effect of ECMO (a previously described risk factor for IC) could be assessed. We subsequently excluded patients who had IC before either of these dates.

A multivariable logistic regression model was used to investigate which risk factors were associated with IC. Variables were selected a priori based on clinical relevance and literature review. Age was evaluated but excluded from the final model due to collinearity with comorbidity burden, which was prioritized given its stronger association with invasive candidiasis and greater clinical relevance.. The odds ratios (ORs) and 95% confidence intervals (CIs) were computed to estimate the strength of the associations among the baseline risk factors on the outcome of IC. Baseline risk factors were captured either on or prior to the index date.

## Mortality analysis

For this analysis, we matched one exposed—defined as a lung transplant recipient (LTR) who developed IC—to two unexposed on age (±5 years), sex, index date with respect to the LT admission date, and length of hospital stay. These variables were selected to control for potential confounders known to influence both the development of IC and post-transplant mortality. Age was used as a matching variable and demonstrated good covariate balance post-matching; therefore, it was not included in the final Cox proportional hazards model. We further assessed the covariate balance between the groups by calculating the standardized differences for each of the covariates in the pre and post matched cohort. A commonly accepted cutoff of 0.10 was used as the threshold for imbalance, where values ranging from 0 to < 0.10 indicate negligible imbalance. All imbalanced variables outside of the ± 0.10 range were controlled for in the final Cox Proportional Hazard (PH) model. The proportional hazards assumption was assessed using Schoenfeld residuals, and no violations were identified.

For the PH model, the first date of IC was the index date. For patients without IC, we do not have a similar date but instead have an LT admission date as their first entrance date. To adjust for this, we first ran an exhaustive match using the LT admission date as an index date. Next, we reset all the exposed groups' index date to the first date of IC after the LT admission date. Then a pseudo index date for the unexposed group was set to be equidistant as the exposed groups' LT admission date to IC, where the minimal difference in the patients' LT length of stay were given precedence in the matching criteria. This allowed us to line up the matching unexposed groups' index dates with the exposed group's index date and help minimize potential confounding in timing to mortality. Enrollment was re-evaluated and all baseline characteristics 90 days prior to the newly assigned index date were analyzed. We then completed a PH model along with a Kaplan Meier curve to evaluate the effect of IC on patient mortality between the groups.

All statistical tests were two-sided, and p-values were reported to quantify the strength of evidence against the null hypothesis. A p-value threshold of <0.05 was used as a general reference for interpreting the results. All analyses were conducted using SAS Enterprise Guide 7.13 (SAS Institute Inc., Cary, NC) and Stata version 16.0 (Stata Corp).

## Results

We identified a total of 2,879 unique patients with LT codes from January 1, 2005, to November 30, 2023; 1,279 patients met inclusion and exclusion criteria (Fig 1). The most common indication for lung transplant was idiopathic pulmonary fibrosis (IPF) followed by non-IPF interstitial lung disease (ILD) and chronic obstructive pulmonary disease (COPD) (Table 1). Most patients underwent bilateral lung transplant. The median follow-up was 469.0 (IQR: 745.0) days. Most LTRs were prescribed

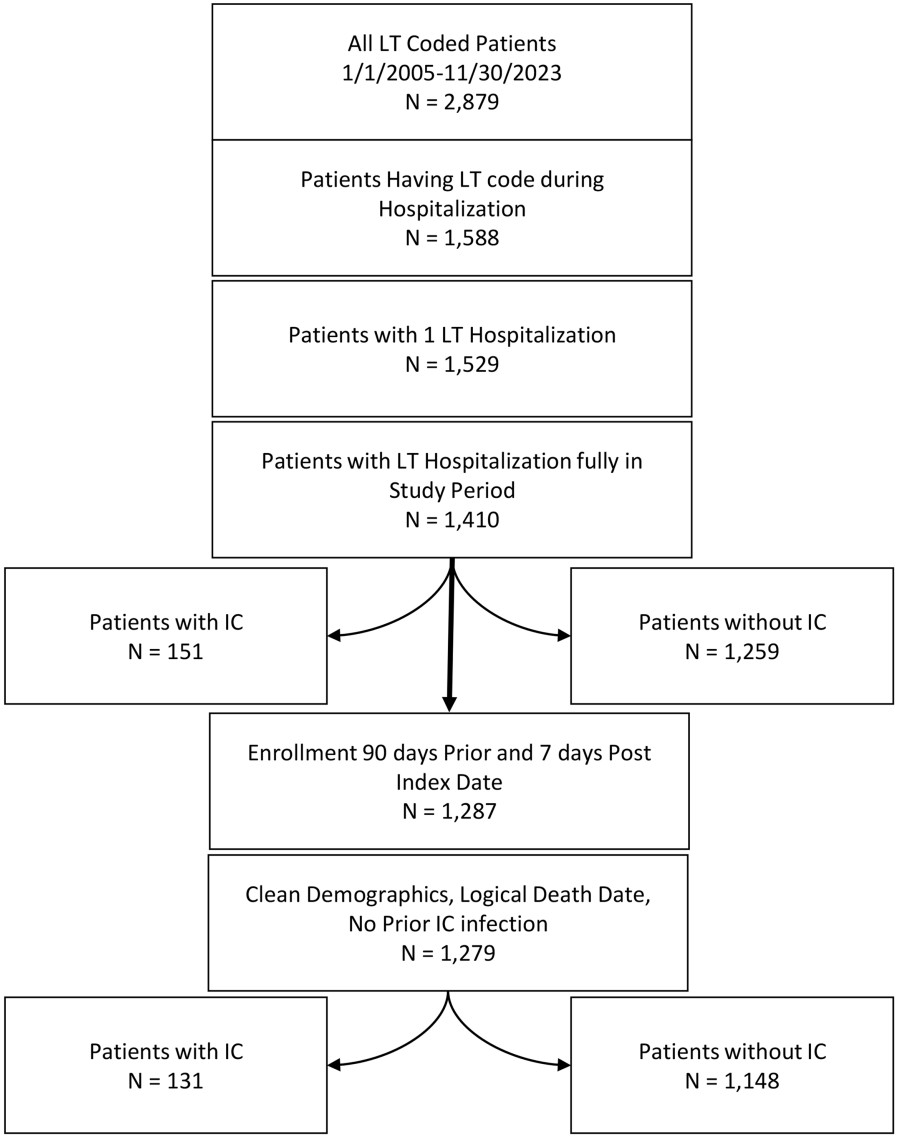

**Fig 1. Flow chart for the development of the final non-matched cohort.**

antifungal medications during their LT admission. The most commonly prescribed antifungal medications were mold-active azoles (voriconazole, posaconazole, and isavuconazonium) followed by nystatin and narrower spectrum azoles (itraconazole and fluconazole).

### Invasive candidiasis

Among the 1,279 LTRs included in the study, 131 (10.2%) developed IC following transplant. The majority of initial episodes of IC occurred during the transplant hospitalization. Median time from LT admission to a diagnostic code for IC was 32.0 (IQR 192.0) days. Based on the diagnostic billing codes, locally invasive candidiasis/ deep-seated candidiasis was most common followed by candidemia. Invasive candidiasis diagnostic codes in administrative claims data are limited in

**Table 1. Non-matched cohort characteristics.**

| | Median (IQR) or Count (%) n = 1,279 |
|---|---|
| Age | 60.0 (52.0, 66.0) |
| Male Sex | 784 (61.3%) |
| Region | |
| Midwest | 340 (26.6%) |
| West | 171 (13.4%) |
| South and Northeast | 768 (60.1%) |
| Race | |
| White | 924 (72.2%) |
| Black | 124 (9.7%) |
| Other/ Unknown | 231 (18.1%) |
| Reason for Transplant | |
| COPD/ Bronchiectasis | 241 (18.8%) |
| Cystic Fibrosis | 110 (8.6%) |
| Idiopathic Pulmonary Fibrosis | 619 (48.4%) |
| Other Interstitial Lung Disease | 260 (20.3%) |
| Other | 49 (3.8%) |
| Type of Transplant | |
| Single | 307 (24.0%) |
| Bilateral | 972 (76.0%) |
| Follow-up | 469.0 (224.0, 969.0) |
| Died during Follow-up | 289 (22.6%) |
| Days to Death from LT | 395.0 (165.0, 991.0) |
| Antifungal Medications During LT Stay | 1,194 (93.4%) |
| Antifungal Medications Prescribed During LT Stay* | |
| Amphotericin (liposomal or deoxycholate) | 50 (3.9%) |
| Itraconazole or Fluconazole | 117 (9.2%) |
| Voriconazole, Posaconazole, or Isavuconazonium | 699 (54.7%) |
| Nystatin | 317 (24.8%) |
| Echinocandin | 166 (13.0%) |
| Developed IC | 131 (10.2%) |
| When IC Occurred (n = 131) | |
| Transplant Hospitalization | 69 (52.7%) |
| After Transplant Hospitalization | 62 (47.3%) |
| Type of IC Based on Diagnostic Code | |
| Locally Invasive/ Deep Seated | 96 (73.3%) |
| Candidemia | 24 (18.3%) |
| Other | 19 (14.5%) |
| Time from LT to IC | 32.0 (0, 192.0) |

COPD- Chronic Obstructive Pulmonary Disease, IC- Invasive Candidiasis, LT- Lung Transplant

*Some patients had more than 1 antifungal medication prescribed

Age is presented in years. Follow-up and Time from LT to IC are presented in days.

Median (IQR) where values within parentheses include the 25th and 75th percentiles.

clinical detail and do not distinguish between specific anatomical sites (e.g., thoracic vs. bloodstream) or severity of infection, which may affect the classification of IC presentations in this study.

Lung transplant recipients who developed IC had longer hospital stays on average (26.0 days, IQR: 53.0 days) compared to those without IC (20.0 days, IQR: 25.0; $p < 0.0001$) (Table 2). They were also more likely to have a re-operation compared to those without IC ($p = 0.003$). Days on ECMO was greater in those with IC (11.0 days, IQR: 41.0) compared to those without IC (7.0 days, IQR: 13.0; $p = 0.010$). Although ECMO use was not different between the groups ($p = 0.411$).

## Risk factors for invasive candidiasis: multivariable analysis

Post-transplant ECMO use of greater than 8 days was associated with approximately a 2.3-fold increase in the odds of developing invasive candidiasis. Other clinical factors, including antifungal prophylaxis in the 90 days prior to IC, CMV disease, diabetes mellitus, high comorbidity burden, bilateral lung transplant, and pre-transplant steroid use, were not meaningfully associated with increased risk. None of the primary indications for lung transplant (e.g., COPD, ILD, IPF) were associated with increased odds of developing invasive candidiasis. Full results are provided in Table 3.

**Table 2. Characteristics of lung transplant recipients who developed invasive candidiasis compared to those who did not.**

| | Invasive Candidiasis Median (IQR) or Count (%) n = 131 | No Invasive Candidiasis Median (IQR) or Count (%) n = 1,148 | *p*-value |
|---|---|---|---|
| Age | 60.0 (51.0, 67.0) | 60.0 (52.0, 66.0) | 0.500 |
| Male Sex | 85 (64.9%) | 699 (60.9%) | 0.371 |
| Reason for Transplant | | | 0.500 |
| COPD/ Bronchiectasis | 23 (17.6%) | 218 (19.0%) | |
| Cystic Fibrosis | <11 (<8.4%) | 100 (8.7%) | |
| Idiopathic Pulmonary Fibrosis | 60 (45.8%) | 559 (48.7%) | |
| Other Interstitial Lung Disease | 35 (26.7%) | 225 (19.6%) | |
| Other | <11 (<8.4%) | 46 (4.0%) | |
| Type of Transplant | | | 0.581 |
| Single | 34 (26.0%) | 273 (23.8%) | |
| Bilateral | 97 (74.1%) | 875 (76.2%) | |
| Follow-up | 517.0 (245.0, 938.0) | 468.0 (220.0, 975.0) | 0.852 |
| Prescribed Corticosteroids Prior to LT Stay | 103 (78.6%) | 881 (76.7%) | 0.631 |
| High Pre-Transplant Co-morbidity Burden[+] | 64 (48.9%) | 590 (51.4%) | 0.582 |
| Pre-Transplant Diabetes Mellitus | 14 (10.7%) | 217 (18.9%) | 0.021 |
| Antifungal Medications During LT Stay | 117 (89.3%) | 1,077 (93.8%) | 0.051 |
| Re-operation During LT Stay | 52 (36.7%) | 313 (27.3%) | 0.003 |
| LT Hospitalization Length of Stay | 26.0 (15.0, 68.0) | 20.0 (13.0, 38.0) | <0.0001 |
| Post-Transplant ECMO | 13 (9.9%) | 90 (7.8%) | 0.411 |
| Days of ECMO Use | 11.0 (9.0, 50.0) | 8.0 (2.0, 15.0) | 0.010 |
| Renal Replacement Therapy | <11 (<8.4%) | 45 (3.9%) | 0.233 |
| CMV Disease Following Transplant | 40 (30.5%) | 293 (25.5%) | 0.221 |

CMV- Cytomegalovirus, COPD- Chronic Obstructive Pulmonary Disease, ECMO- Extra-corporeal Membrane Oxygenation, LT- Lung Transplant

[+]Defined Elixhauser Comorbid Conditions score of 4 or higher.

Age is presented in years. Follow-up and LT hospitalization length of stay are presented in days.

Median (IQR) where values within parentheses include the 25th and 75th percentiles.

**Table 3. Multivariable logistic regression for risk factors for invasive candidiasis following lung transplant.**

| Observations = 1,279 | OR | *p-value* | 95% CI | |
|---|---|---|---|---|
| **Gender** (Ref. Female) | | | | |
| Male Sex | 1.15 | 0.491 | 0.78 | 1.70 |
| **Transplant Reason** (Ref. COPD/ Bronchiectasis) | | | | |
| Cystic Fibrosis | 0.70 | 0.450 | 0.28 | 1.74 |
| Idiopathic pulmonary fibrosis | 1.01 | 0.971 | 0.58 | 1.77 |
| Other Interstitial Lung Disease | 1.79 | 0.061 | 0.98 | 3.29 |
| Other | 0.55 | 0.500 | 0.10 | 3.08 |
| **Type of Transplant** (Ref. Single) | | | | |
| Bilateral Transplant | 0.90 | 0.643 | 0.59 | 1.39 |
| **ECMO Days** (Ref. ≤ 8) | | | | |
| 9+ | 2.34 | 0.04* | 1.03 | 5.34 |
| **Elixhauser** (Ref. ≤ 3) | | | | |
| 4+ | 0.86 | 0.670 | 0.44 | 1.70 |
| **Baseline Risk Factors** | | | | |
| CMV disease | 1.91 | 0.395 | 0.43 | 8.51 |
| Pre-Transplant Steroid Use | 1.12 | 0.641 | 0.69 | 1.83 |
| Antifungal Prescription | 1.49 | 0.165 | 0.85 | 2.59 |
| Diabetes Mellitus | 1.77 | 0.156 | 0.80 | 3.92 |
| Renal Failure | 0.75 | 0.520 | 0.31 | 1.80 |

CI- Confidence Interval, COPD- Chronic Obstructive Pulmonary Disease, CMV- Cytomegalovirus, OR- Odds Ratio, Ref.- Reference

### Effect of invasive candidiasis on mortality

To evaluate the impact of invasive candidiasis on survival, we matched 126 lung transplant recipients with IC to 252 without IC (Fig 2). As shown in Table 4, baseline characteristics between recipients with and without IC were well balanced after matching, with standardized differences falling below the commonly accepted threshold of 0.1, indicating good covariate balance across groups. The median follow-up after the index date was 439.0 days (IQR: 770.0 days), with 97 deaths (25.7%) observed during the follow-up period. Median time to death was 347.0 days (IQR: 703.0 days). Lung transplant recipients with IC had a higher mortality rate compared to matched recipients without IC, with an adjusted hazard ratio of 2.31 (95% CI: 1.45 to 3.67), reflecting more than a twofold increase in the risk of death.

Table 5 presents the results of the Cox proportional hazards model evaluating the impact of IC on all-cause mortality. All-cause mortality was higher in those who developed IC (event rate per 100 person-years: 11.32; HR: 2.13; 95% CI 1.45 to 3.12, $p < 0.001$). This held true when matching IC exposed to unexposed (event rate per 100 person-years: 12.87; HR: 2.31; 95% CI 1.45 to 3.67, $p < 0.001$) (Fig 3).

### Discussion

Invasive candidiasis occurred in approximately 10.2% of LTRs in our national cohort, with a median time to diagnosis of 32 days. This finding is consistent with a prior single-center study by Baker et al., which reported an IC prevalence of 11.4% and a similar median time to diagnosis of 31 days [7]. In contrast, Marinelli et al. described a lower prevalence of 3.5%; however, their study only evaluated episodes of candidemia and did not include other forms of invasive Candida infections, such as surgical site or pleural space infections [8]. This difference highlights the importance of case definition in determining incidence estimates. Our study used administrative billing codes, which allow for large-scale cohort identification but may include both candidemia and deep-seated non-bloodstream infections. The broader definition used in

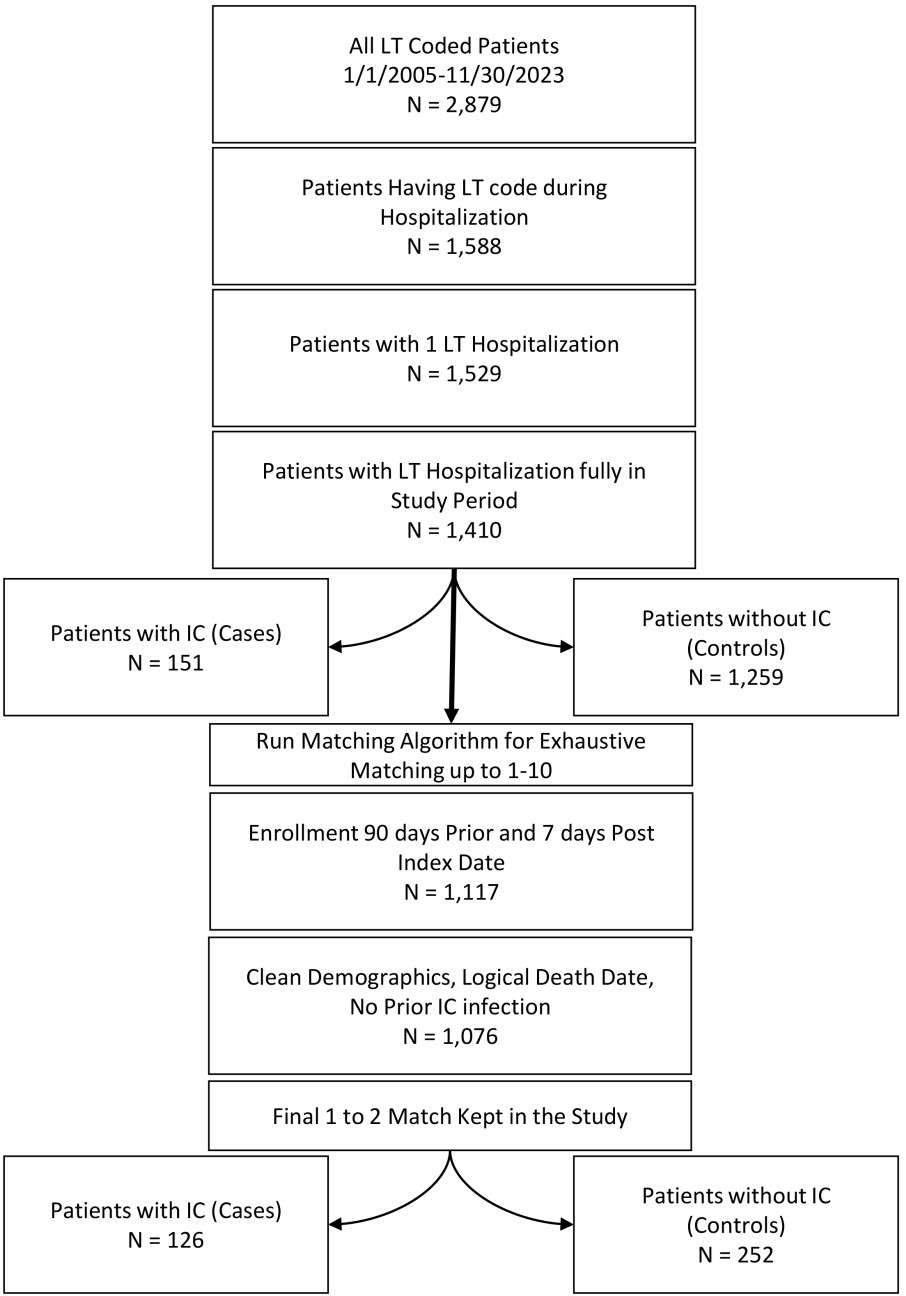

**Fig 2. Flow chart for development of the matched cohort of lung transplant recipients with invasive candidiasis matched to age, sex, and length of hospital stay controls (1 with invasive candidiasis to 2 without invasive candidiasis).**

our claims-based approach may explain the higher observed prevalence compared to studies that only included microbiologically confirmed bloodstream infections. Furthermore, differences in antifungal prophylaxis practices—such as the use of inhaled amphotericin B versus systemic triazoles—may have contributed to variability in observed rates across studies. These findings align with prior large-scale surveillance data from the Transplant-Associated Infection Surveillance Network (TRANSNET), which identified Candida species as the most common cause of invasive fungal infections across

**Table 4. Matched cohort of lung transplant recipients with invasive candidiasis matched to age, sex, and length of hospital stay controls (1 invasive candidiasis to 2 without invasive candidiasis).**

| | Invasive Candidiasis Median (IQR) or Count (%) n=126 | No Invasive Candidiasis Median (IQR) or Count (%) n=252 | Standardized Difference after match |
|---|---|---|---|
| Age | 60.0 (52.0, 67.0) | 60.0 (52.0, 66.0) | 0.01 |
| Male Sex | 81 (64.3%) | 162 (64.3%) | 0.00 |
| Reason for Transplant | | | 0.32* |
| COPD/ Bronchiectasis | >21 (>16.7%) | 218 (19.0%) | |
| Cystic Fibrosis | <11 (<8.7%) | >33 (>13.1%) | |
| Idiopathic Pulmonary Fibrosis | 61 (48.4%) | 105 (41.7%) | |
| Other Interstitial Lung Disease | 33 (26.2%) | 60 (26.8%) | |
| Other | <11 (<8.7%) | <22 (8.4%) | |
| Prescribed Corticosteroids Prior to LT Stay | 108 (85.7%) | 219 (86.9%) | 0.03 |
| Pre-Transplant Diabetes Mellitus | 37 (29.4%) | 73 (29.0%) | 0.01 |
| Antifungal Medications During LT Stay | 99 (78.6%) | 212 (84.1%) | 0.11* |
| CMV Disease Following Transplant | 39 (30.1%) | 59 (23.4%) | 0.12* |

*All imbalanced variables outside of the ±0.10 range were controlled for in the final Cox Proportional Hazard (PH) model.

Age is presented in years.

Median (IQR) where values within parentheses include the 25th and 75th percentiles.

Per OptumLabs® data use policies, cell counts ≤11 are masked and reported using "<" or ">" symbols to protect patient privacy.

solid organ transplant recipients [3]. Our findings build on this by focusing specifically on lung transplant recipients and demonstrating that IC remains a common and clinically important complication, even in the context of widespread antifungal exposure.

We found that post-transplant ECMO support of greater than 8 days is a risk factor for IC in LTRs. Post-transplant ECMO support and renal replacement therapy were risk factors reported by Marinelli et. al.[8] and coincides with risk factors reported in other critically ill populations [11,21]. Re-operation has not previously been reported as a risk factor for IC in LTRs. However, it is a known risk factor for IFI in other solid organ transplant recipients, and an open chest has been reported as a potential risk factor for IC in heart transplant recipients [22,23]. In our data, due to both the small sample size and the need to show a clear chronology directionality with potential risk factors, we were not able to fully explore re-operation within the index hospitalization or post-transplant renal replacement therapy as risk factors for IC. While re-operation was higher in the IC group, we cannot be certain if that was a risk factor for or consequence of IC. We saw no difference in the incidence of post-transplant renal replacement therapy between the groups. We decided to focus on ECMO support as a potential risk factor because of its established association with invasive fungal infections in transplant and critically ill populations, and because the timing of ECMO initiation provided a reliable anchor to establish chronological directionality between exposure and development of IC. This maintained a 90-day washout period of IC and a proper baseline period to collect risk factors of contracting IC post-index date. While the other measures are important, the direction of causality would be less reliable in the model if they were present.

Our dataset does not include information on ECMO cannulation sites. While femoral access may increase the risk of invasive candidiasis due to higher potential for contamination and vascular complications, further studies are needed to confirm this association.

Absence of antifungal prophylaxis during the index hospitalization was not a risk factor for IC in our cohort, but around 90% of patients were prescribed some type of antifungal medication making it difficult to discern if absence of an

**Table 5. Cox proportional hazards model evaluating the impact of IC on all-cause mortality.**

| Cox Accounting for remaining unbalanced covariates after the match | | | | | |
|---|---|---|---|---|---|
| | Hazard Ratio | Std. Err. | *p-value* | 95% CI | |
| **Description** | | | | | |
| Invasive Candidiasis | 2.31 | 0.56 | < 0.0001 | 1.45 | 3.67 |
| **Region code (Cont. Midwest)** | | | | | |
| South/ Northeast | 1.00 | 0.26 | 0.987 | 0.60 | 1.67 |
| West | 0.71 | 0.27 | 0.372 | 0.34 | 1.50 |
| **Race code (Cont. Black)** | | | | | |
| Unknown | 2.14 | 1.09 | 0.136 | 0.79 | 5.83 |
| White | 1.56 | 0.76 | 0.363 | 0.60 | 4.07 |
| **Transplant Reasion** | | | | | |
| COPD or Bronchiectasis | 0.98 | 0.47 | 0.963 | 0.39 | 2.48 |
| IPF | 1.10 | 0.39 | 0.788 | 0.55 | 2.21 |
| Non-IPF ILD | 1.26 | 0.47 | 0.531 | 0.61 | 2.62 |
| Other | 1.45 | 0.95 | 0.577 | 0.40 | 5.27 |
| Unknown | 4.64 | 4.73 | 0.133 | 0.63 | 34.27 |
| **Baseline Risk Factors** | | | | | |
| CMV Disease | 2.15 | 0.80 | 0.038 | 1.04 | 4.45 |
| Rx Antifungal Use | 0.98 | 0.23 | 0.915 | 0.62 | 1.54 |
| **Baseline Elixhauser Components** | | | | | |
| Cardiac Arrhythmia | 0.65 | 0.17 | 0.099 | 0.39 | 1.08 |
| Deficiency Anemia | 0.86 | 0.50 | 0.8 | 0.27 | 2.71 |
| Chronic Pulmonary Disease | 1.02 | 0.26 | 0.952 | 0.61 | 1.69 |
| Coagulopathy | 1.71 | 0.48 | 0.057 | 0.99 | 2.98 |
| Fluid and Electrolyte Disorders | 0.75 | 0.20 | 0.274 | 0.45 | 1.25 |
| Hypertension without Complications | 1.04 | 0.24 | 0.873 | 0.66 | 1.63 |
| Hypothroidism | 0.86 | 0.38 | 0.73 | 0.36 | 2.04 |
| Liver Disease | 1.71 | 0.51 | 0.073 | 0.95 | 3.08 |
| Obesity | 1.62 | 0.56 | 0.162 | 0.82 | 3.20 |
| Other Neurological Disorders | 1.97 | 0.61 | 0.031 | 1.07 | 3.63 |
| Renal Failure | 1.07 | 0.32 | 0.821 | 0.60 | 1.91 |
| Valvular Disease | 1.05 | 0.28 | 0.852 | 0.62 | 1.77 |

CI- Confidence Interval, COPD- Chronic Obstructive Pulmonary Disease, CMV- Cytomegalovirus, ILD- Interstitial Lung Disease, IPF- Idiopathic Pulmonary Fibrosis

antifungal medication would increase the risk of IC. Given that over 90% of patients received antifungal prophylaxis, our ability to detect a protective effect was likely limited by the lack of variability in exposure, and the absence of observed benefit should be interpreted with caution.

Traditional risk factors for IC in non-immunosuppressed patients such as diabetes mellitus and co-morbidity burden were not associated with increased risk of IC in our cohort. Likewise, CMV disease has been implicated as a risk factor for IFI [23], but this was not a risk factor for IC in our cohort. These differences are possibly due to the small sample size or possibly because other studies have grouped IC and invasive mold infections as a composite outcome. Pathophysiologically, CMV as a risk factor for invasive mold infections makes sense as it causes direct tissue damage within the lungs weakening alveolar barrier defenses [24].

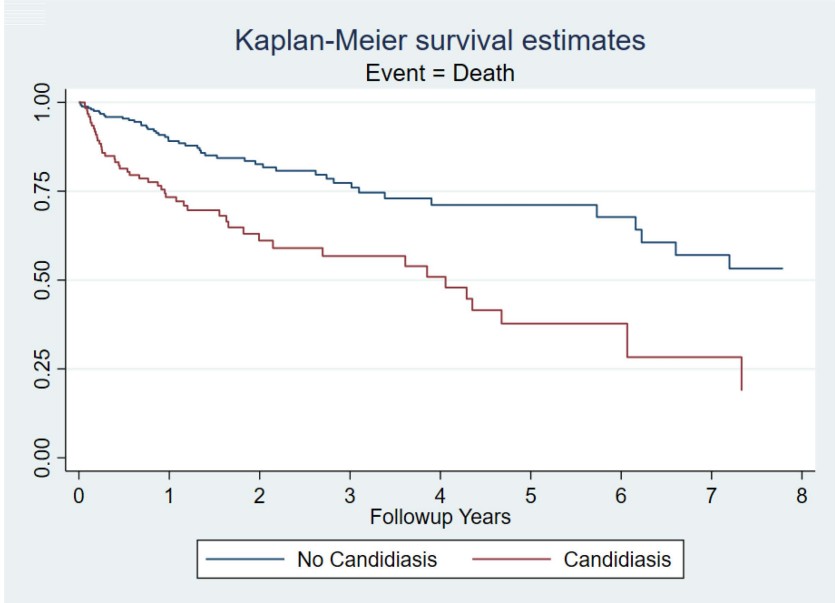

**Fig 3. Kaplan-Meier survival curve comparing lung transplant recipients with and without invasive candidiasis (IC).** The x-axis represents time from index date (in years), and the y-axis represents the probability of survival. The curves reflect unadjusted survival estimates. The effect of IC on all-cause mortality was further assessed using a Cox proportional hazards model, adjusted for residual covariate imbalance following matching (adjusted HR: 2.31; 95% CI: 1.45–3.67), as reported in the Results section.

We found that lung transplant recipients with invasive candidiasis had a more than twofold increased risk of death compared to matched recipients without IC, consistent with the adjusted hazard ratio of 2.31 reported in our Cox proportional hazards model. Marinelli et. al. found in their single center cohort study that LTRs with candidemia had an increased 30-day post-transplant mortality [8]. A 3-fold increase in mortality in the first year following transplant has also been described in thoracic transplant recipients at Stanford University between 1980 and 2004 [25]. While we have had significant advancements in antifungal medications since the Stanford University cohort, mortality rates in LTRs with IC remains staggeringly high. Future efforts should be focused on preventative measures for the development of IC in LTRs. This needs to go beyond pharmacologic prophylaxis, as most patients in our cohort were on some type of antifungal medication and the incidence of IC was around 10%.

While our findings align with prior single-center studies, confirmation in other cohorts—particularly those with different antifungal prophylaxis practices or transplant protocols—will be important to validate these results and assess generalizability. Our findings do have practical implications for clinical management and resource allocation in LT care. Identifying prolonged ECMO support as a risk factor for IC highlights a high-risk subgroup where heightened surveillance or targeted antifungal strategies may be warranted. Additionally, the observed association between IC and increased mortality underscores the need for early recognition and timely intervention. The data also inform antifungal stewardship efforts by emphasizing that universal prophylaxis may not fully prevent IC and should be balanced against drug-related toxicities. As such, these results support a risk-based approach to IC prevention and may help guide clinical decision-making in the early post-transplant period.

## Limitations

The primary constraint inherent to any observational study is the potential for uncontrolled confounding variables. In our investigation, due to the absence of pertinent data within the de-identified cohort, we were unable to consider additional

risk factors such as intensified immunosuppressive regimens, pre-transplant colonization status, or parenteral nutrition. Nevertheless, we diligently managed known confounding factors by including variables such as sex, age, race/ethnicity, geographical region, year of transplant, primary diagnosis, type of transplant (single versus double or heart/lung), and comorbidities in our analysis.

We acknowledge the potential for selection bias introduced by excluding patients with a history of invasive candidiasis prior to transplant while retaining those with prior antifungal exposure, which may reflect underlying differences in base-line health status. Additionally, a slightly higher exclusion rate among patients who later developed IC (13% vs. 8%) could suggest unmeasured pre-transplant differences that we were unable to fully control for.

One notable advantage afforded by the OLDW is the real-world, enriched racial, ethnic, and geographic diversity of the cohort it encompasses, setting it apart from single observational studies. In comparison to frequently used datasets like the Scientific Registry of Transplant Recipients (SRTR), although our lung transplant cohort is relatively smaller, it furnishes additional data not readily available in SRTR. However, it's imperative to acknowledge a drawback associated with this dataset; it is confined to patients enrolled in commercial and Medicare Advantage health plans. This limitation may potentially curtail the generalizability of our study, as it does not encompass uninsured patients or those enrolled in other government health plans.

Furthermore, the inherent nature of the de-identified dataset introduces certain limitations. For instance, our reliance on the Social Security Death Master File for mortality data comes with constraints, notably the limitations of this database since 2011, which may not capture up to one-third of deaths [26]. To mitigate this limitation, we supplemented the Social Security Death Master File with information derived from discharge status and insurance discontinuation due to death (insurance knowledge of death), particularly since the majority of early LT deaths occur within the hospital setting. While we acknowledge the possibility of missing a small proportion of patients who may have passed away outside the hospital environment, we anticipate that this phenomenon would be comparable between the groups we are comparing. Lastly, it is important to recognize that we employed billing codes to identify IC. These codes are contingent on accurate diagnosis and coding by the treating healthcare provider, introducing a potential source of variability and error in our analysis.

We dichotomized ECMO duration and comorbidity burden for ease of interpretation and consistency with prior studies, but acknowledge this approach may have limited our ability to detect dose-response relationships. Our analytic approach to evaluating post-IC mortality involved assigning matched unexposed LTRs a pseudo-index date aligned with the timing of IC diagnosis in the exposed group. While this allowed for comparison of outcomes from similar post-transplant timepoints, it may introduce bias if the assumptions underlying index date alignment are not fully met. Modeling IC as a time-dependent covariate in a Cox model represents an alternative approach and may help validate our findings in future studies.

## Conclusion

In our multi-institutional cohort, IC has a prevalence of around 10% in LTRs with most episodes initially occurring during the index hospitalization. Post-transplant ECMO support for greater than 8 days increases the risk of IC, and patients with IC have more re-operations and longer hospital length of stay. LTRs who developed IC had a two-fold increased risk of death.

## Author contributions

**Conceptualization:** Kelly Pennington, Herb Heien, Steve G. Peters, Patricio Escalante, Raymund R. Razonable, Cassie C. Kennedy.

**Data curation:** Kelly Pennington, Herb Heien, Xiaoxi Yao, Patricio Escalante, Che Ngufor, Raymund R. Razonable.

**Formal analysis:** Kelly Pennington, Herb Heien, Hemang Yadav, Xiaoxi Yao, Patricio Escalante, Che Ngufor, Raymund R. Razonable, Cassie C. Kennedy.

**Investigation:** Kelly Pennington, Bradley White, Che Ngufor, Raymund R. Razonable.

**Methodology:** Kelly Pennington, Herb Heien, Hemang Yadav, Xiaoxi Yao, Bradley White, Steve G. Peters, Che Ngufor, Raymund R. Razonable, Cassie C. Kennedy.

**Resources:** Kelly Pennington, Che Ngufor, Cassie C. Kennedy.

**Software:** Che Ngufor.

**Supervision:** Hemang Yadav, Xiaoxi Yao, Steve G. Peters, Patricio Escalante, Che Ngufor, Cassie C. Kennedy.

**Validation:** Steve G. Peters.

**Writing – original draft:** Kelly Pennington.

**Writing – review & editing:** Herb Heien, Hemang Yadav, Xiaoxi Yao, Bradley White, Steve G. Peters, Patricio Escalante, Che Ngufor, Raymund R. Razonable, Cassie C. Kennedy.

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
