## [Decision Letter · Decision Letter 0]

25 Mar 2025

PONE-D-25-06214Invasive Candidiasis Following Lung Transplant: An Assessment of Impact Utilizing a National Insurance Claims CohortPLOS ONE

Dear Dr. Pennington,

Thank you for submitting your manuscript to PLOS ONE. After careful consideration, we feel that it has merit but does not fully meet PLOS ONE’s publication criteria as it currently stands. Therefore, we invite you to submit a revised version of the manuscript that addresses the points raised during the review process. Please submit your revised manuscript by May 09 2025 11:59PM. If you will need more time than this to complete your revisions, please reply to this message or contact the journal office at plosone@plos.org . Please include the following items when submitting your revised manuscript:

We look forward to receiving your revised manuscript.

Kind regards,

*
**Ali Amanati**
*

**Academic Editor**

PLOS ONE

4. Please include a copy of Table 1-4 which you refer to in your text on page 6.

Additional Editor Comments:

Dear Authors,

Your manuscript [PONE-D-25-06214] has passed the review stage and is ready for ‎revision. ‎

To ensure the Editor and Reviewers can recommend that your revised manuscript be ‎accepted, ‎‎‎please pay careful attention to each comment posted underneath ‎this email. This way we ‎can ‎‎avoid future clarifications and revisions, moving swiftly to ‎a decision.‎

Technical points:‎

‎1. Please provide a point-by-point response to the Editor and reviewer's comments

‎2. Please highlight all the amends on your manuscript with a yellow color

‎3. Use line numbering and page number in the next submission‎

‎4. Improve the English language of the manuscript‎

Reviewers' comments:

Reviewer's Responses to Questions

**Comments to the Author**

1. Is the manuscript technically sound, and do the data support the conclusions?

Reviewer #1: Yes

Reviewer #2: Partly

Reviewer #3: Yes

Reviewer #4: Partly

Reviewer #5: Yes

Reviewer #6: Yes

Reviewer #7: Yes

2. Has the statistical analysis been performed appropriately and rigorously? 

Reviewer #1: Yes

Reviewer #2: I Don't Know

Reviewer #3: Yes

Reviewer #4: I Don't Know

Reviewer #5: Yes

Reviewer #6: Yes

Reviewer #7: Yes

3. Have the authors made all data underlying the findings in their manuscript fully available?

Reviewer #1: Yes

Reviewer #2: Yes

Reviewer #3: Yes

Reviewer #4: No

Reviewer #5: No

Reviewer #6: No

Reviewer #7: Yes

4. Is the manuscript presented in an intelligible fashion and written in standard English?

Reviewer #1: Yes

Reviewer #2: Yes

Reviewer #3: Yes

Reviewer #4: No

Reviewer #5: Yes

Reviewer #6: Yes

Reviewer #7: Yes

5. Review Comments to the Author

Reviewer #1: - Please don’t use abbreviates in the abstract (ECMO).

- Introduction lines 6 and 7 are wrong. Please see this article: Mayer FL, Wilson D, Hube B. Candida albicans pathogenicity mechanisms. Virulence. 2013 Feb 15;4(2):119-28. doi: 10.4161/viru.22913. Epub 2013 Jan 9. PMID: 23302789; PMCID: PMC3654610.

- Lines 10-11 according to reference 7 is not true.

- Please define invasive candidiasis “In our population, IC was defined as any diagnostic code for IC (ICD-9: 112.4, 112.5, 112.8, 112.4, 112.83, 112.5;5 ICD-10: B37.1, B37.5, B37.6, B37.7) in any position following the LT procedure”

Reviewer #2: The paper presented is a very interesting retrospective analysis of the risk factors leading to Candidiasis infection after lung transplant. The following issues regarding the statistical considerations should be addressed prior to publication:

1) There appears to be selection bias in that patients with a prior history of IC were excluded, but patients with a prior exposure to anti-fungals were not. Furthermore, based on Figure 1 it appears 13% of IC cases but only 8% of non-IC cases were excluded. This suggests possible selection bias.

2) It is unclear what the exact selection methods were for variable choices in the multivariable regression models. Based on the language Page 5 Lines 11-28, many factors were considered. However Table 3 reports the OR for only a select set of variables. For example why was age not included in the risk analysis? Similar issue for PH regression.

3) For the PH model of mortality, it is unclear why the analysis was complicated so with the challenge of the index date for cases and controls. Why not use the index date as LT for both cases and controls, but use date of IC as a time-dependent covariate? The choice to manipulate the index date for controls opens up potential bias in the results. The overall statistical question is asking does an IC infection post LT impact overall mortality. Give that the median time from LT admission to IC was 32d (IQR 0 to 192) suggests possible confounding of analysis and interpretation of the results.

4) Page 6, Lines 49-50: As over 90% of subjects had antifungal prophylactic therapy it is not surprising there is no observed benefit.

5) Page 6, Lines 47-54: this paragraph is a reiteration of the data in Table 3.

6) Why are the results of the PH model not shown as a table? The HR in Figure 3 should be that from the PH model, and the legend should clarify this.

7) Given this is a retrospective look at over 18 years of data, it would be useful to report on the annual rate of infections (maybe over 3 year windows?) and test if the overall rate of infection decreased over the years.

Minor issues:

1) Abstract indicates data was collected January 1, 2005, to July 31, 2023 while Page 4 Lines 49-50 indicate data study period was January 1, 2005 to December 31, 2023. Suggest abstract should be revised.

2) Revise Page 5 Line 43 to read more clearly: “Analysis of both risk factors in the development of IC and mortality from IC required us to use two different models and extraction methods.”

3) Table 1 and Table 2 should be combined with the first column reporting cohort total, second column IC, third column non-IC.

4) Table 3 can exclude the Z-score value. Also “Control” is an improper term, use “Reference” instead.

Reviewer #3: (abstract) Write the aim of the study at past tense.

(abstract) Define ECMO abbreviation.

(general comment) Do not start a sentence with an abbreviation.

(abstract)Define "index hospitalization".

(introduction) Please be specific when you say "at high risk".

(introduction) Define early in "early post-lung transplant period"

(introduction) Please provide the frequencies for "antifungal medications can have significant adverse effects".

(introduction) Write the aim of the study at past tense.

(methods) Who can access OLDW? It is open access?

(methods) "Re-transplantations were excluded" why?

(methods) Detail the method use to collect mortality data.

(methods) Detail the Elixhauser comorbidity score.

(methods) "compared using a t test" even those with deviations from the theoretical normal distribution?

(methods) Describe the methods used to decide which variables entered in multivariable model.

(methods) "we matched one case" the case is defined as patient with IF? Please clarify.

(methods) Tell the readers the level of significance used in the statistical analysis.

(results) Define axes and units of measurements on Kaplan-Meier graph.

(results) Table 1 - 4: do not include the units of measurements in the body of the table. Generally, when round brackets are used the lower and upper values are not included in the range; please clarify.

(results) Report p-values with at least 3 decimals

(discussion) "Our study utilized administrative claims data to evaluate both risk factors and the impact of IC in a diverse cohort of LTRs." This is duplicated information and should be deleted.

(discussion) Are your results expected to be confirm on other cohorts?

(discussion) Discuss the practical utility of the reported results.

Reviewer #4: This article does not mention diagnostic methods. How is pulmonary candidiasis diagnosed? The medications taken by the patient were not mentioned in this article. How much did their use contribute to the disease's development? What is the diagnostic code for IC?

Reviewer #5: Congratulations to this mostly very well written manuscript. It has been one of very few manuscripts that I reviewed during the last few years that didn't have any major statistical flaws. The only reason that I selected major instead of minor revision is the number of questions that I have and that should be clarified before this manuscript is published. I hope they are helpful:

1. The authors write in the Statistical Analysis section that in the descriptive tables, they show mean + sd or median and IQR as appropriate. However, in Tables 1 and 2 there only is the median/IQR shown. Is the reason for this that all variables seem to be nonnormally distributed? Or just for was of presentation (to avoid confusions due to switching between mean and median)? In any case, only the median/IQR should be mentioned in the methods.

2. Follow-up question from the above: As far as I understood, all tests performed in Table 2 are t-tests? But if the (or some of the) data are nonnormally distributed, a nonparametric test should be used. Please change that.

3. Please delete the stars denoting statistical significance from the tables. Nowadays, there is a strong trend towards quantifying the evidence against the null hypothesis instead of making a binary cutoff to distinguish "significant" from "non-significant". I don't mind you discussing significance in the text (although I clearly prefer writing something like "There was no evidence for a difference"), but do not use stars in the tables.

4. How were the covariates for the multivariable logistic regression model chosen? I don't fully understand the line "We then assessed a priori which risk factors were associated." How did you do that? Is that a fancy way of writing "We asked experts which risk factors to include"? Please clarify.

By the way, I very much like that you obviously didn't choose the covariates based on their p-value or something wrong like that. But some information how you actually chose them would be very helpful.

5. I recommend to use the word "Reference category" instead of "Control" in Table 3. Control can be misunderstood as being a control group, especially as you also have matching in your manuscript.

6. Both the ECMO days and the Elixhauser variable were dichotomized, but for example in Table 2 the Elixhauser is included as count variable. I don't like dichotomizing very much because you usually loose information. Have you tried to include them in a continuous way? If so, what was the result? Have you looked at the descriptives if it might be reasonable to include them in a non-linear way? And if that is not an option (with a good reason): How were the thresholds chosen? If there is a clinical reasoning behind that (if for example the threshold is used in clinical practice) this should be stated and a reference for this should be given. If the threshold was selected by the authors, please also write how and why you did that.

7. Two hazard ratios are mentioned in the text. First, 2.13, then 2.31 "when matching IC cases to controls", and the latter one is also printed below the Kaplan-Meier curve. I don't understand which model the first value comes from. Is that a simple, unmatched model with just one influential variable? I also find it irritating that the value 2.31 comes from the adjusted model that the authors mention, but that it is printed below the KM curve suggests to me that it comes from an unadjusted model. Please clarify. Think about not showing the HR below the curve, as the HR comes from an adjusted model.

8. Please print the results of the final Cox model in an additional table. I think it important to also see what hazard ratios the other influential variables show.

9. Did you check the proportional hazards assumption? Please mention that and the results of this.

10. I don't understand the > and < signs in Table 4. Please explain.

Reviewer #6: Overall

I read with great interest the article entitled “Invasive Candidiasis Following Lung Transplant: An Assessment of Impact Utilizing a National Insurance Claims Cohort”, which falls within the aim of this journal. This study aimed to assess the incidence, risk factors, and impact of invasive candidiasis on mortality in lung transplant recipients using administrative claims data from individuals enrolled in commercial and Medicare Advantage health plans in the US.

In my honest opinion, the topic and results are interesting for audiences and the paper is almost well structured enough to attract the readers’ attention. However, the authors should consider and clarify some points and improve the paper, as suggested below [*: major points, #: minor points].

Abstract section

# Please note that the tone of your study’s objective in the Introduction part should align with the verb tense used in the Methods and Results parts. Therefore, instead of “This study aims to…”, consider using ‘This study aimed to….’.

# Please ensure that the final sentence of the Results part clarifies whether the finding is statistically significant. Therefore, include the relevant statistical measures.

* Writing the Conclusion part is crucial, but it is currently insufficient. Additionally, the first sentence is very similar to the first line of the previous part. So please revise it. You may use the following suggestion or a similar one: ‘Invasive candidiasis affects approximately 10% of lung transplant recipients and is linked to higher mortality, prolonged hospitalization, and increased surgical interventions. These findings underscore the importance of early identification and targeted preventive strategies to improve post-transplant outcomes’.

Introduction section

* Please note that the phrase “Since Candida spp. do not have the ability to digest and invade tissues” is not entirely accurate based on various studies, including this one: https://doi.org/10.1016/j.rsma.2023.103258. So kindly revise it.

# Please note that the phrase “its effectiveness has not been established in the lung transplant population” is stated too definitively. It would be preferable to rephrase it as ‘evidence exists but is inconclusive’.

# Please note that here “One tool that has been utilized to assess clinical incidence and risk factor questions in other clinical domains are large claims databases, such as OptumLabs® Data Warehouse”, “One tool” does not align with “are”. It is recommended to restructure the sentence. A suggested revision is: ‘Large claims databases, such as OptumLabs® Data Warehouse (OLDW), have been widely used to assess disease incidence and risk factors across various clinical domains’.

Methods section

* Regarding your sentence: “We required subjects to have at least 90 days of continuous health plan coverage prior to their lung transplant date and at least 30 days post”. Please clarify the rationale behind selecting the 90-day pre-transplant and 30-day post-transplant timeframes.

# In the part Outcomes of Interest, please note that ICD code 112.5 appears to be duplicated.

* Since your study follows a cohort design, it is preferable not to use the terms “Case” and “Control”, which are more relevant to case-control studies. Please revise these terms throughout the text and figures, replacing them with ‘diseased’ and ‘non-diseased’, respectively. Additionally, when referring to mortality outcomes, please use ‘exposed’ and ‘unexposed’ instead.

# In the Mortality Analysis part, it would be preferable to briefly explain the rationale behind matching for the selected variables. For instance for confounding variables.

Results section

# Please ensure consistency in punctuation. In this instance, (619; 48.4%) uses a semicolon, whereas a comma would be more appropriate for consistency.

* Please note that (Q1, Q3) is not equivalent to the interquartile range (IQR), as IQR is calculated as Q3 minus Q1. It is recommended to report IQR as a single value in the text while maintaining the (Q1, Q3) format in tables for clarity.

# Regarding the sentence “The most commonly prescribed antifungal were mold-active azoles”, please note that “antifungal” should be written as ‘antifungals’ or, preferably, ‘Antifungal Medications’ to maintain consistency with the terminology used in the tables.

# In the first paragraph of this section, please remove all instances of (n, x%) as this information is already provided in the table.

# Please revise the sentence “Of these, 131 (10.2%) LTRs developed IC following LT”. to explicitly refer to ‘lung transplant recipients’ instead of using “Of these”.

* Preferably, ensure that the variable titles used in the tables are consistent with those in the main text. For example, term like “locally invasive candidiasis/deep-seated candidiasis” in the text are somewhat ambiguous, whereas the table clearly refers to “Pulmonary Candidiasis”. Please make the necessary revisions and include any additional explanations in parentheses.

# Please clarify what you mean by “IC diagnostic codes are limited”.

* In the Invasive Candidiasis part, please remove all instances of (n, x%) since this information is already provided in the tables. Additionally, mentioning other details is unnecessary; instead, simply state that all had a P-value < 0.05 (also please clarify that if you have considered this threshold in the Statistical Analysis part). Please apply this recommendation to other parts as well.

# For the sentence: “Post-transplant ECMO (OR: 2.34; 95% CI 1.03 to 5.34, p = 0.043) use of greater than 8 days was the only risk factor significantly associated with post-transplant IC on multivariable modeling”. Please remove the information inside the parentheses as previously suggested. Additionally, after “associated with”, add ‘was associated with around 2.3-fold increase in’ to clarify the magnitude of the association.

# Please clarify this phrase “No single indication for transplant increased the odds of IC”.

# Please note that the first two sentences in the Effect of Invasive Candidiasis on Mortality part are not appropriate for this part.

* Please ensure that Table 4 is adequately interpreted, as its interpretation is even more important than that of Table 3.

* Please provide appropriate titles for both figures, as they currently lack titles.

# Regarding Table 3, please note that it is unnecessary to include SE and Z values. It is recommended to remove these values from the table.

* Ensure that consistent variable names are used across all tables and carefully review them for uniformity. Additionally, clarify the rationale for including certain variables in Table 4 that are absent from Table 3 (e.g., COPD/Bronchiectasis).

Discussion section

* Please note that this section currently requires further elaboration from literature. It is recommended to include a more detailed comparison of your study with previous research and, if possible, please discuss the reasons for any discrepancies, which could be attributed to for example, different methodologies.

* Regarding the sentence: “We decided to focus on ECMO support as a potential risk factor by setting the index date ...”, please clarify the rationale for selecting ECMO support.

# Please note that the sentence “We cannot glean ECMO configuration placement position from our dataset. It is likely that femoral lines portend a higher risk of IC than ...” presents a claim without sufficient scientific support. It is recommended to revise and integrate the following phrase: ‘Our dataset does not include information on ECMO cannulation sites. While femoral access may increase the risk of IC, further studies are needed to confirm this association’.

* Regarding your sentence: “We found that patients with IC have more than twice the probability of mortality compared to their matched partners”. I did not find the corresponding results in the previous section.

Good Luck,

Reviewer #7: A very comprehensive study. Data are available, so it is a nice addition to the potential resources made available to others researching similar topics. It is a "complex read" but that is very much essential here. I noted a misspell [Invasive Candidiasis. . . . 1 paragraph, last line, "diagnositic"(sic)] -- (Can't believe I saw it however, due to the 'density' of this writing). I like the detail oriented approach even for the following: implementation of Elixhauser over Charlton Comorbidity Scoring--that makes sense with, provides and example for, a study focused on Labs data.

6. PLOS authors have the option to publish the peer review history of their article (what does this mean? ). If published, this will include your full peer review and any attached files.

**Do you want your identity to be public for this peer review?** For information about this choice, including consent withdrawal, please see our Privacy Policy .

Reviewer #1: **Yes: ** Parisa Badiee

Reviewer #2: No

Reviewer #3: No

Reviewer #4: No

Reviewer #5: No

Reviewer #6: No

Reviewer #7: **Yes: ** Brian L Altonen

---

## [Author Response · Author response to Decision Letter 1]

10 May 2025

Dear Editorial Team:

We thank the editorial team and all reviewers for their time and detailed evaluation of our manuscript, “Invasive Candidiasis Following Lung Transplant: An Assessment of Impact Utilizing a National Insurance Claims Cohort.” We are especially grateful for the thoughtful critiques, which have significantly strengthened the clarity, methodological transparency, and overall quality of the manuscript. In the pages that follow, we provide detailed responses to each reviewer comment. All revisions made to the manuscript are noted with corresponding page and line numbers. We believe these changes enhance the rigor and impact of our work and sincerely appreciate the opportunity to revise and resubmit.

Reviewer #1: 

We thank Reviewer #1 for their thoughtful review and valuable feedback. We are pleased that you found the topic of invasive candidiasis following lung transplantation to be of interest. Your comments helped us clarify key points and improve the overall quality of the manuscript. Our response to individual comments are outlined below:

- Please don’t use abbreviates in the abstract (ECMO).

Thank you. This has been written in long form with abbreviation parenthetically given the commonality of the term ECMO in the healthcare profession.

- Introduction lines 6 and 7 are wrong. Please see this article: Mayer FL, Wilson D, Hube B. Candida albicans pathogenicity mechanisms. Virulence. 2013 Feb 15;4(2):119-28. doi: 10.4161/viru.22913. Epub 2013 Jan 9. PMID: 23302789; PMCID: PMC3654610.

Thank you for this important correction and for pointing us to the relevant literature. We agree that our original statement was inaccurate. We have revised the sentence to better reflect the known pathogenic mechanisms of Candida species, including their ability to adhere to and invade host tissues. The revised text on page 4, lines5-8 now reads:

“Candida spp. possess a range of virulence factors that facilitate epithelial adhesion, invasion, and tissue damage, contributing to pathogenesis in the setting of impaired host immunity or barrier disruption.”

- Lines 10-11 according to reference 7 is not true.

We thank the reviewer for this important clarification. Upon re-reviewing Reference 7, we agree that it does not support the broad statement regarding breakthrough IC with nebulized AmBisome. We have revised the sentence to remove the overstatement and better reflect the cited literature. The revised text on page 4, lines 9-11 now reads:

“While many lung transplant centers in the United States employ antifungal prophylaxis targeting mold infections(10), variation exists in the agents used and the duration of prophylaxis. Some studies have suggested that the use of inhaled amphotericin B without systemic antifungal prophylaxis may be insufficient to prevent all forms of invasive fungal infection, including invasive candidiasis (7).”

- Please define invasive candidiasis “In our population, IC was defined as any diagnostic code for IC (ICD-9: 112.4, 112.5, 112.8, 112.4, 112.83, 112.5;5 ICD-10: B37.1, B37.5, B37.6, B37.7) in any position following the LT procedure”

We appreciate the reviewer’s request for clarification. In our study, invasive candidiasis (IC) was defined using a set of International Classification of Diseases (ICD-9 and ICD-10) diagnostic codes intended to capture clinically significant invasive infections. We have added the following to Page 5, Lnes 6-9: “These codes include candidemia (e.g., 112.5, B37.7), disseminated candidiasis (e.g., 112.5, 112.83), and organ-specific invasive infections (e.g., B37.5 for candidal peritonitis and B37.6 for candidal endocarditis). We acknowledge that claims data may not distinguish between confirmed invasive disease and coding inaccuracies.”

Reviewer #2: 

The paper presented is a very interesting retrospective analysis of the risk factors leading to Candidiasis infection after lung transplant. The following issues regarding the statistical considerations should be addressed prior to publication:

We thank Reviewer #2 for their thoughtful and constructive feedback. We are pleased that you found the study of interest. We appreciate your detailed review of the statistical methodology and have addressed each of your points below to enhance the clarity and rigor of our analysis.

1) There appears to be selection bias in that patients with a prior history of IC were excluded, but patients with a prior exposure to anti-fungals were not. Furthermore, based on Figure 1 it appears 13% of IC cases but only 8% of non-IC cases were excluded. This suggests possible selection bias.

We appreciate the reviewer’s observation regarding potential selection bias. Patients with a prior diagnosis of invasive candidiasis (IC) were excluded to ensure that all identified IC events occurred de novo after lung transplantation, allowing for more accurate assessment of post-transplant risk factors and outcomes. In contrast, prior antifungal exposure was not used as an exclusion criterion, as antifungals may be prescribed for a variety of prophylactic or therapeutic indications unrelated to IC and do not necessarily indicate prior infection.

With respect to the exclusion discrepancy noted in Figure 1, we thank the reviewer for highlighting this detail. The slightly higher exclusion rate among patients who later developed IC (13% vs. 8%) likely reflects the fact that patients with IC may have had more complex clinical courses or prolonged pre-transplant hospitalizations, during which time IC was diagnosed and coded. While this introduces a possibility of selection bias, we have added a statement acknowledging this as a limitation on page 8, line 8-11: “We acknowledge the potential for selection bias introduced by excluding patients with a history of invasive candidiasis prior to transplant while retaining those with prior antifungal exposure, which may reflect underlying differences in baseline health status. Additionally, a slightly higher exclusion rate among patients who later developed IC (13% vs. 8%) could suggest unmeasured pre-transplant differences that we were unable to fully control for.”

2) It is unclear what the exact selection methods were for variable choices in the multivariable regression models. Based on the language Page 5 Lines 11-28, many factors were considered. However Table 3 reports the OR for only a select set of variables. For example why was age not included in the risk analysis? Similar issue for PH regression.

Thank you for this insightful comment. We apologize for the lack of clarity regarding our variable selection process for the multivariable models. In the logistic regression model assessing risk factors for IC, we included a priori selected clinical variables based on prior literature and clinical plausibility. These included transplant-related factors (e.g., ECMO use, re-operation), demographic characteristics (e.g., sex, region), and comorbidities. Although age was considered, it was ultimately excluded from the final model due to lack of significant univariable association and collinearity with other factors such as comorbidity burden. We have now clarified this in the Methods section on page 6, lines 3-5: “Variables were selected a priori based on clinical relevance and literature review. Age was evaluated but excluded from the final model due to collinearity with comorbidity burden and lack of a significant univariable association with IC.”

For the Cox proportional hazards model, we used matched cohorts (on age ±5 years, sex, index date relative to transplant, and hospital length of stay), and the final model included only variables that remained imbalanced after matching. Since age was a matching variable and showed good covariate balance (standardized difference <0.1), it was not included in the final PH model. We have added text to clarify this approach in the Methods on page 6, lines 11-12: “Age was used as a matching variable (±5 years) and demonstrated good covariate balance post-matching; therefore, it was not included in the final Cox proportional hazards model.”

3) For the PH model of mortality, it is unclear why the analysis was complicated so with the challenge of the index date for cases and controls. Why not use the index date as LT for both cases and controls, but use date of IC as a time-dependent covariate? The choice to manipulate the index date for controls opens up potential bias in the results. The overall statistical question is asking does an IC infection post LT impact overall mortality. Give that the median time from LT admission to IC was 32d (IQR 0 to 192) suggests possible confounding of analysis and interpretation of the results.

We thank the reviewer for this important comment and agree that modeling IC as a time-dependent covariate is a valid alternative approach. Our goal, however, was to assess mortality following an episode of IC, and to compare outcomes in those with and without IC who were at a similar timepoint post-transplant and had similar post-operative trajectories (as reflected by hospital length of stay and index date proximity). To do this, we aligned the timing of follow-up between cases and controls by anchoring the index date for each control to mirror the case's time from transplant to IC diagnosis, thus mitigating immortal time bias and facilitating interpretation of post-IC outcomes.

We recognize that this approach may introduce its own limitations, particularly with regard to assumptions around matching and event timing. We have acknowledged this as a limitation in the Discussion page 8, lines 34-38: “Our analytic approach to evaluating post-IC mortality involved assigning matched controls a pseudo-index date aligned with the timing of IC diagnosis in cases. While this allowed for comparison of outcomes from similar post-transplant timepoints, it may introduce bias if the assumptions underlying index date alignment are not fully met. Modeling IC as a time-dependent covariate in a Cox model represents an alternative approach and may help validate our findings in future studies.”

4) Page 6, Lines 49-50: As over 90% of subjects had antifungal prophylactic therapy it is not surprising there is no observed benefit.

We agree with the reviewer that the high prevalence of antifungal prophylaxis in our cohort likely limited our ability to detect a protective association. This high background rate reduces variability in exposure, potentially obscuring any true effect of prophylaxis on invasive candidiasis (IC) risk. We have added language to the Discussion on page 7, lines 44-48: “Given that over 90% of patients received antifungal prophylaxis, our ability to detect a protective effect was likely limited by the lack of variability in exposure, and the absence of observed benefit should be interpreted with caution.”

5) Page 6, Lines 47-54: this paragraph is a reiteration of the data in Table 3.

We appreciate the reviewer’s perspective. While the content in this paragraph does overlap with Table 3, we believe that summarizing key findings from the multivariable analysis in the text improves accessibility and supports interpretation for readers who may not focus on tables alone. These results help contextualize the lack of association for several anticipated risk factors, such as CMV disease and diabetes mellitus, and emphasize that only prolonged ECMO support was statistically significant. We have reviewed the paragraph to ensure it remains concise and non-redundant and believe its inclusion strengthens the narrative presentation of our findings. Edited text is on page 6, lines 54-60: “Post-transplant ECMO (OR: 2.34; 95% CI 1.03 to 5.34, p =0.043) use of greater than 8 days was the only risk factor significantly associated with post- transplant IC on multivariable modeling. Other clinical factors, including the presence of antifungal prophylaxis during the 90 days prior to IC (OR: 1.49; 95% CI 0.85 to 2.59, p=0.165), CMV disease (OR: 1.91; 95% CI 0.43 to 8.51, p = 0.395), diabetes mellitus (OR: 1.77; 95% CI 0.80 to 3.92, p=0.156), high co-morbidity burden (OR: 0.86; 95% CI 0.44 to 1.70, p=0.67), bilateral lung transplant (OR: 0.90; 95% CI 0.59 to 1.39, p=0.643), and pre-transplant steroid use (OR: 1.12; 95% CI 0.69, 1.83, p=0.641) were not significant risk factors for IC. No single indication for transplant increased the odds of IC.

6) Why are the results of the PH model not shown as a table? The HR in Figure 3 should be that from the PH model, and the legend should clarify this.

We appreciate the reviewer’s suggestion to enhance clarity. We have added a new table (now Table 5) summarizing the results of the Cox proportional hazards model, including the hazard ratio, 95% confidence interval, and p-value. This allows for clearer interpretation and complements the Kaplan-Meier curve presented in Figure 3. We have also updated the Figure 3 legend to specify that the hazard ratio shown is derived from the Cox model.

Figure 3 Legend now states, “Kaplan-Meier survival curve comparing lung transplant recipients with and without invasive candidiasis (IC). The hazard ratio (HR: 2.31; 95% CI: 1.45–3.67) is derived from the Cox proportional hazards model adjusted for variables with residual imbalance following matching.”

Results section on page 7, lines 7-10 now states, “Table 5 presents the results of the Cox proportional hazards model evaluating the impact of IC on all-cause mortality. All-cause mortality was significantly higher in those who developed IC (event rate per 100 person-years: 11.32; HR: 2.13; 95% CI 1.45 to 3.12, p<0.001). This held true when matching IC cases to controls (event rate per 100 person-years: 12.87; HR: 2.31; 95% CI 1.45 to 3.67, p<0.001) (Figure 3).

Minor issues:

1) Abstract indicates data was collected January 1, 2005, to July 31, 2023 while Page 4 Lines 49-50 indicate data study period was January 1, 2005 to December 31, 2023. Suggest abstract should be revised.

Thank you for identifying this inconsistency. We have revised the abstract to reflect the correct study period of January 1, 2005, to December 31, 2023, in alignment with the Methods section.

2) Revise Page 5 Line 43 to read more clearly: “Analysis of both risk factors in the development of IC and mortality from IC required us to use two different models and extraction methods.”

Thank you. We have updated this statementon on page 5, line 47-48.

3) Table 1 and Table 2 should be combined with the first column reporting cohort total, second column IC, third column non-IC.

We appreciate the reviewer’s suggestion. We considered combining Tables 1 and 2 but felt that presenting baseline characteristics (Table 1) and post-transplant outcomes (Table 2) separately improves readability and keeps the focus clear for each table. Combining them would result in a substantially larger and denser table, potentially making it more difficult for readers to interpret key findings. We have retained the current format to preserve clarity and emphasize the distinction between baseline variables and outcome data.

4) Table 3 can exclude the Z-score value. Also “Control” is an improper term, use “Reference” instead.

Thank you for this helpful suggestion. We have removed the Z-score column from Table 3 and replaced the term “Control” with “Reference” to more appropriately indicate the referent category in the regression model.

Reviewer #3: 

We thank Reviewer #3 for their thoughtful review and constructive feedback. We appreciate your time and expertise in evaluating our manuscript and have carefully addressed each of your comments to improve the clarity, accuracy, and overall quality of the work.

(abstract) Write the aim of the study at past tense.

This has been updated on page 3, lines 3-4: “This study aimed to assess the incidence, risk factors, and impact of IC on mortality in LTRs using a national insurance claims cohort.”

(abstract) Define ECMO abbreviation.

This has been done at the suggestion of Reviewer 1.

(general comment) Do not start a sentence with an abbreviation.

Thank you for this ed

---

## [Decision Letter · Decision Letter 1]

14 Jun 2025

PONE-D-25-06214R1Invasive Candidiasis Following Lung Transplant: An Assessment of Impact Utilizing a National Insurance Claims CohortPLOS ONE

Dear Dr. Pennington,

Thank you for submitting your manuscript to PLOS ONE. After careful consideration, we feel that it has merit but does not fully meet PLOS ONE’s publication criteria as it currently stands. Therefore, we invite you to submit a revised version of the manuscript that addresses the points raised during the review process. Please submit your revised manuscript by Jul 29 2025 11:59PM. If you will need more time than this to complete your revisions, please reply to this message or contact the journal office at plosone@plos.org . Please include the following items when submitting your revised manuscript:

We look forward to receiving your revised manuscript.

Kind regards,

**
*Ali Amanati*
**

Academic Editor

PLOS ONE

Journal Requirements:

Additional Editor Comments:

Dear authors, ‎

‎The invited reviewers posted new comments. So, the manuscripts ‎‎require a ‎round of revision.‎ Please provide a point-by-point response to the ‎‎reviewer's ‎comments and highlight all the ‎amends on your manuscript with ‎‎yellow color.‎ ‎

Yours,

Reviewers' comments:

Reviewer's Responses to Questions

**Comments to the Author**

1. If the authors have adequately addressed your comments raised in a previous round of review and you feel that this manuscript is now acceptable for publication, you may indicate that here to bypass the “Comments to the Author” section, enter your conflict of interest statement in the “Confidential to Editor” section, and submit your "Accept" recommendation.

Reviewer #1: All comments have been addressed

Reviewer #2: All comments have been addressed

Reviewer #4: All comments have been addressed

Reviewer #5: (No Response)

Reviewer #6: (No Response)

2. Is the manuscript technically sound, and do the data support the conclusions?

Reviewer #1: Yes

Reviewer #2: Yes

Reviewer #4: Yes

Reviewer #5: Yes

Reviewer #6: Yes

3. Has the statistical analysis been performed appropriately and rigorously? 

Reviewer #1: Yes

Reviewer #2: Yes

Reviewer #4: I Don't Know

Reviewer #5: Yes

Reviewer #6: Yes

4. Have the authors made all data underlying the findings in their manuscript fully available?

Reviewer #1: Yes

Reviewer #2: Yes

Reviewer #4: No

Reviewer #5: No

Reviewer #6: Yes

5. Is the manuscript presented in an intelligible fashion and written in standard English?

Reviewer #1: Yes

Reviewer #2: Yes

Reviewer #4: Yes

Reviewer #5: Yes

Reviewer #6: Yes

6. Review Comments to the Author

Reviewer #1: The manuscript reviewed by 6 reviewer and revised as my previous comments. It is suitable for publication.

Reviewer #2: Thank you for addressing my comments. Although I would prefer to see the mortality modeled with IC as a time-dependent covariate, the choice of a psudo index date is not unheard of in statistical modeling. The manuscript is much improved with better clarity.

Reviewer #4: The authors present a well-conducted and clinically relevant retrospective cohort study that addresses a critical and underexplored aspect of post-lung transplantation care. The use of a large national insurance claims database provides robust real-world evidence on the incidence and impact of invasive candidiasis (IC) in lung transplant recipients. The methodology is sound. The manuscript is clearly written, the results are compelling, and the conclusions are well-supported by the data. I recommend acceptance of this article.

Reviewer #5: Thank you for answering all my questions and changing the manuscript accordingly. There is one last thing that should be formulated differently:

The authors explain (both in "my" answer and in one for reviewer 2) that they chose the variables for the regression model based on clinical knowledge and literature, but they state that age was excluded "due to collinearity with other factors and lack of a significant univariable association". But exactly significance or non-significance in a univariable model should not be used to decide a bout including a variable, because the p-value can change considerably as soon as additional variables are taken into account.

Therefore, the authors should delete this part of the sentence. If there indeed is strong (multi-)collinearity, this is reason enough to not include age. If, however, the collinearity is not very strong, I suggest to include age in the model as an additional covariate. In any case, it has to be explored and explained properly.

Reviewer #6: I sincerely thank the authors for their thoughtful and thorough revisions to the manuscript. Almost all of my previous comments have been addressed. Before resubmitting, I recommend reviewing the revised version without Track Changes to ensure there are no minor issues, such as the presence of two periods at the end of the conclusion sentence in the abstract. Also, please note that the code “112.4” appears twice in the following sentence: “In our population, ... for IC (ICD-9: 112.4, 112.5, 112.8, 112.4, 112.83; ICD-10: 16 B37.1, B37.5, B37.6, B37.7) in ... procedure”.

Additionally, I appreciate your attention to comment “Since your study follows a cohort design, it is preferable not to use the terms “Case” and “Control”, which are more relevant to case-control studies. Please revise these terms throughout the text and figures, replacing them with ‘diseased’ and ‘non-diseased’, respectively. Additionally, when referring to mortality outcomes, please use ‘exposed’ and ‘unexposed’ instead”. However, a few instances of these terms still appear in the text and should preferably be corrected.

Good Luck,

7. PLOS authors have the option to publish the peer review history of their article (what does this mean? ). If published, this will include your full peer review and any attached files.

**Do you want your identity to be public for this peer review?** For information about this choice, including consent withdrawal, please see our Privacy Policy .

Reviewer #1: **Yes: ** Parisa Badiee

Reviewer #2: No

Reviewer #4: No

Reviewer #5: No

Reviewer #6: No

---

## [Author Response · Author response to Decision Letter 2]

18 Jul 2025

Dear Editorial Team:

We thank the editorial team and all reviewers for their time and detailed evaluation of our manuscript, “Invasive Candidiasis Following Lung Transplant: An Assessment of Impact Utilizing a National Insurance Claims Cohort.”

Reviewer #1: 

The manuscript reviewed by 6 reviewer and revised as my previous comments. It is suitable for publication.

We sincerely thank the reviewer for their positive feedback and continued support of our manuscript.

Reviewer #2: 

Thank you for addressing my comments. Although I would prefer to see the mortality modeled with IC as a time-dependent covariate, the choice of a psudo index date is not unheard of in statistical modeling. The manuscript is much improved with better clarity.

We appreciate the reviewer’s thoughtful feedback and are grateful for the recognition of the improvements made to the manuscript. We acknowledge that modeling IC as a time-dependent covariate is a valid and elegant alternative. In this analysis, we selected a pseudo index date approach to align with our dataset structure and facilitate matching on temporal proximity and hospital length of stay. We agree that future work could benefit from incorporating time-dependent modeling to validate and extend our findings.

Reviewer #4: 

The authors present a well-conducted and clinically relevant retrospective cohort study that addresses a critical and underexplored aspect of post-lung transplantation care. The use of a large national insurance claims database provides robust real-world evidence on the incidence and impact of invasive candidiasis (IC) in lung transplant recipients. The methodology is sound. The manuscript is clearly written, the results are compelling, and the conclusions are well-supported by the data. I recommend acceptance of this article.

We thank the reviewer for their positive feedback and thoughtful assessment of our study. We greatly appreciate the recommendation for acceptance.

Reviewer #5: 

Thank you for answering all my questions and changing the manuscript accordingly. There is one last thing that should be formulated differently:

The authors explain (both in "my" answer and in one for reviewer 2) that they chose the variables for the regression model based on clinical knowledge and literature, but they state that age was excluded "due to collinearity with other factors and lack of a significant univariable association". But exactly significance or non-significance in a univariable model should not be used to decide a bout including a variable, because the p-value can change considerably as soon as additional variables are taken into account.

Therefore, the authors should delete this part of the sentence. If there indeed is strong (multi-)collinearity, this is reason enough to not include age. If, however, the collinearity is not very strong, I suggest to include age in the model as an additional covariate. In any case, it has to be explored and explained properly.

We thank the reviewer for this important and insightful comment. We agree that variable selection should not rely on statistical significance in univariable models, and we appreciate the opportunity to clarify our rationale.

In our analysis, age demonstrated moderate collinearity with comorbidity burden (Elixhauser score), and including both in the multivariable model introduced instability in the model estimates. Given that comorbidity burden was more strongly associated with our outcome of interest and is clinically relevant in this context, we prioritized its inclusion over age. We have revised the manuscript to reflect this clarification and have removed the reference to univariable significance. The updated text now reads:

“Age was evaluated but excluded from the final model due to collinearity with comorbidity burden, which was prioritized given its stronger association with IC and greater clinical relevance.”

Reviewer #6:

 I sincerely thank the authors for their thoughtful and thorough revisions to the manuscript. Almost all of my previous comments have been addressed. Before resubmitting, I recommend reviewing the revised version without Track Changes to ensure there are no minor issues, such as the presence of two periods at the end of the conclusion sentence in the abstract.

We thank the reviewer for their careful review and kind comments. We have removed the extraneous period in the abstract and carefully reviewed the revised manuscript for any remaining formatting or typographical errors before resubmission.

Also, please note that the code “112.4” appears twice in the following sentence: “In our population, ... for IC (ICD-9: 112.4, 112.5, 112.8, 112.4, 112.83; ICD-10: 16 B37.1, B37.5, B37.6, B37.7) in ... procedure”.

This has been corrected in the revised version, and the duplicate 112.4 code has been removed.

Additionally, I appreciate your attention to comment “Since your study follows a cohort design, it is preferable not to use the terms “Case” and “Control”, which are more relevant to case-control studies. Please revise these terms throughout the text and figures, replacing them with ‘diseased’ and ‘non-diseased’, respectively. Additionally, when referring to mortality outcomes, please use ‘exposed’ and ‘unexposed’ instead”. However, a few instances of these terms still appear in the text and should preferably be corrected.

Thank you for this important clarification and for previously highlighting the need to align terminology with cohort study design conventions. We carefully reviewed the full manuscript and have now revised all remaining instances of the terms “case” and “control.” Specifically, we have replaced “case”/“control” with “diseased”/“non-diseased” in sections referring to IC status, and with “exposed”/“unexposed” when referring to matched groups in the mortality analysis. Figure titles and flowchart labels have been updated accordingly. We appreciate the reviewer’s close reading and attention to methodological precision.

Once again, we would like to express our gratitude to the reviewers for their valuable feedback, which has been instrumental in refining our manuscript. Thank you for your consideration and for the opportunity to resubmit our work.

Sincrerely,

Kelly M. Pennington, MD

---

## [Editor Report · Decision Letter 2]

29 Jul 2025

Invasive Candidiasis Following Lung Transplant: An Assessment of Impact Utilizing a National Insurance Claims Cohort

PONE-D-25-06214R2

Dear Dr. Kelly Pennington,

We’re pleased to inform you that your manuscript has been judged scientifically suitable for publication and will be formally accepted for publication once it meets all outstanding technical requirements.

Kind regards,

*
**Ali Amanati**
*

**Academic Editor**

PLOS ONE

Additional Editor Comments (optional):

The authors have effectively utilized all available resources and data to enhance the manuscript, making it ‎more scientifically robust than before. Therefore, based on my opinion and the esteemed ‎reviewers' ‎‎comments, it could be published in its current form.‎

Yours‎,‎

---

## [Editor Report · Acceptance letter]

PONE-D-25-06214R2

PLOS ONE

Dear Dr. Pennington,

I'm pleased to inform you that your manuscript has been deemed suitable for publication in PLOS ONE. Congratulations! Your manuscript is now being handed over to our production team.

Kind regards,

on behalf of

Professor Ali Amanati

Academic Editor

PLOS ONE